# *TOR1AIP1*-Associated Nuclear Envelopathies

**DOI:** 10.3390/ijms24086911

**Published:** 2023-04-07

**Authors:** Laurane Mackels, Xincheng Liu, Gisèle Bonne, Laurent Servais

**Affiliations:** 1MDUK Oxford Neuromuscular Center, Department of Paediatrics, University of Oxford, Oxford OX3 9DU, UK; 2Adult Neurology Department, Citadelle Hospital, 4000 Liège, Belgium; 3Sorbonne University, INSERM, Institut de Myologie, Centre de Recherche en Myologie, 75013 Paris, France; 4Neuromuscular Center, Division of Paediatrics, University Hospital of Liège, University of Liège, 4000 Liège, Belgium; 5MDUK Oxford Neuromuscular Center, Department of Paediatrics, NIHR Oxford Biomedical Research Centre, University of Oxford, Oxford OX3 9DU, UK

**Keywords:** lamina-associated protein 1, LAP1, lamina-associated polypeptide 1B, LAP1B, lamina-associated polypeptide 1C, LAP1C, TOR1AIP1, inner nuclear membrane, mutations, clinical spectrum

## Abstract

Human *TOR1AIP1* encodes LAP1, a nuclear envelope protein expressed in most human tissues, which has been linked to various biological processes and human diseases. The clinical spectrum of diseases related to mutations in *TOR1AIP1* is broad, including muscular dystrophy, congenital myasthenic syndrome, cardiomyopathy, and multisystemic disease with or without progeroid features. Although rare, these recessively inherited disorders often lead to early death or considerable functional impairment. Developing a better understanding of the roles of LAP1 and mutant *TOR1AIP1*-associated phenotypes is paramount to allow therapeutic development. To facilitate further studies, this review provides an overview of the known interactions of LAP1 and summarizes the evidence for the function of this protein in human health. We then review the mutations in the *TOR1AIP1* gene and the clinical and pathological characteristics of subjects with these mutations. Lastly, we discuss challenges to be addressed in the future.

## 1. Introduction

Lamina-associated protein 1 (LAP1), encoded by the *TOR1AIP1* gene (OMIM #614512), is a ubiquitously expressed nuclear protein located on the inner nuclear membrane [1,2]. Several isoforms of LAP1 have been reported in various species, but only two have been identified in humans, LAP1B and LAP1C. Although its exact biological functions remain incompletely understood, LAP1 has been linked to a large variety of both ubiquitous and tissue-specific cellular processes in animals and humans, as detailed in this review. Variants in *TOR1AIP1* and their associate phenotypes have been described in several case reports and case series, with a total of 23 affected patients reported worldwide [3,4,5,6,7,8,9,10,11,12]. The aims of this review are to summarize what is known regarding the functions of LAP1 in human health and disease, and to provide a comprehensive overview of *TOR1AIP1*-related disorders. Additionally, we discuss phenotype–genotype associations and highlight points warranting consideration in future studies.

## 2. Methods and Search Strategy

We performed a scoping review through searches of the MEDLINE (using Ovid and PubMed) and Scopus databases. Selected keywords (Appendix A) were combined to create search strategies with adjustments specific to each database as described in the Appendix A. The searches were limited to French and English languages and covered the year 1988 onward; LAP1 was first identified in rat liver cells in 1988 [1]. As our aim was to obtain a holistic overview rather than to answer specific questions, materials and research produced by organizations outside traditional commercial or academic publishing were included. We screened references of selected articles to find relevant work that were not identified through the primary research strategies. Lastly, more focused searches were also conducted for subcategories of LAP1 mechanisms and interactions.

## 3. LAP1 an Integral Protein of the Inner Nuclear Membrane

### 3.1. TOR1AIP1 Encodes Different LAP1 Isoforms

LAP1 is a chromatin-binding integral protein localized at the inner nuclear membrane. It was first discovered in 1988 from a sample of rat liver and was initially identified as an integral inner nuclear membrane protein associated with lamins [1]. This initial study identified three isoforms with molecular masses of 75 kDa, 68 kDa, and 55 kDa, that were later named LAP1A, LAP1B, and LAP1C. These isoforms result from alternative splicing of the *TOR1AIP1* pre-mRNA.

In humans, two LAP1 isoforms have been identified, LAP1B and LAP1C [2,13], which are generated from the *TOR1AIP1* gene due to alternative start site usage [13] (Figure 1). LAP1B is a 584-amino-acid-long protein with a molecular weight of 66.3 kDa [2]. An additional splice variant of LAP1B has also been reported that differs by a CAG deletion in exon 3 [14]. LAP1C lacks 122 amino acids that are present at the N-terminus of LAP1B. LAP1C is 462 amino acids in length and has a molecular weight of 56 kDa [13]. Human LAP1B and LAP1C have molecular weights similar to those of LAP1 isoforms identified in rat and mouse. Of note, LAP1C from animals and humans was shown to be more easily solubilized in vitro than LAP1B when using detergent. This is thought to be due to LAP1C’s shorter nucleoplasmic domain [1,13,15]. The physiological implication of this finding remains unclear.

LAP1 isoforms are type 2 transmembrane proteins that have a single transmembrane domain and that lack N-terminal signal sequences [16]. Structurally, both isoforms have three distinct domains: (1) an N-terminal domain located on the nucleoplasmic side of the inner nuclear membrane, (2) a single transmembrane domain, and (3) a luminal C-terminus domain located in the perinuclear space [2,16,17] (Figure 1). Both LAP1B and LAP1C are subject to post-translational modification including reversible protein phosphorylation, dephosphorylation and methionine oxidation [13,18]. The exact functions of such post-transcriptional changes in relation to human health and diseases have not been fully elucidated.

The functions of the two isoforms have not been completely characterized. Human cells contain a greater amount of LAP1C as compared to LAP1B. The levels of each isoform vary across human tissues, resulting in organ-specific LAP1B/LAB1C ratios [13]. For instance, LAB1B is expressed at higher levels in the brain, heart, and liver, whereas LAP1C is more abundant in the lung, kidney, and spleen. Comparable amount of both isoforms are found in the pancreas, testis, and ovary [13]. Additionally, levels of both isoforms depend on cell maturation status [19,20]. Specifically, higher levels of LAP1C are detected in undifferentiated human cell lines, whereas LAP1B is expressed at higher levels in highly differentiated tissues such as the brain and heart [13,20].

### 3.2. Interactions of LAP1 with Other Proteins in the Nuclear Envelope

The nuclear envelope is a double-lipid-bilayer membrane barrier that surrounds the eukaryotic nucleus and separates the chromosomes from the cytoplasm. The nuclear envelope consists of inner and outer nuclear membranes that are separated by a perinuclear space spanned by nuclear pore complexes [21,22,23] The nuclear lamina, mainly consisting of A- and B-type lamins, lies beneath the inner nuclear membrane within the nucleoplasm. The nuclear lamina ensures structural support to the nuclear membranes and provides anchoring sites for chromatin helping to organize the genome [22,24,25]. Approximatively 80 transmembrane proteins have been found on the inner nuclear membrane [26]. These proteins bind to the nuclear lamina and chromatin, forming a complex web in the nucleoplasmic space. The integrity of the nuclear proteins and their complex interactions are essential to the maintenance and functions of the nucleus [22]. A review of the LAP1 interactome identified 41 potential interactions, 36 of which were confirmed in humans [27]. Since then, LAP1 has been identified in numerous interactome studies in humans, but very few of these were further investigated with regards to their physiological significance. The interactions of LAP1 with lamin A/C and lamin B, torsinA, emerin, PP1, and TRF2 are the best characterized and are discussed in the sections below (Figure 2). The reader can refer to interaction databases and previous reviews for an exhaustive list of interactions involving LAP1 reported so far.

#### 3.2.1. Lamins

Lamins are type V intermediate filament proteins that act as subunits of the nuclear lamina meshwork [28,29]. Lamins ensure maintenance of the nuclear structure and, through impacts on chromatin organization, play a role in gene expression [30,31]. Mutations in *LMNA*, which encodes lamins A and C, are known to cause a broad spectrum of diseases including dilated cardiomyopathy sometimes associated with muscular dystrophy, familial partial lipodystrophy, an autosomal recessive form of Charcot–Marie–Tooth, restrictive dermopathy, bone mandibuloacral dysplasia, and accelerated aging/progeroid syndromes such as Hutchinson Gilford progeria syndrome and atypical Werner syndrome [32,33]. Mutations in *LMNB1*, which encodes lamin B1, lead to a broad range of diseases including autosomal dominant leukodystrophy, ataxia–telangiectasia, and neural tube defects [32,33,34,35]. Lamins were identified as LAP1-interacting proteins [15]. Lamins A, C, and B1 bind to the N-terminal domain of LAP1 in the nucleoplasm, thereby connecting the nuclear lamina to the inner nuclear membrane [1,15,36,37]. The interaction between LAP1 and lamin proteins helps to maintain nuclear envelope structure and chromatin positioning [15,36,38]. Reciprocally, LAP1 ensures appropriate positioning of A-type lamins at the inner nuclear membrane [38].

#### 3.2.2. Emerin

Emerin is an integral protein of the inner nuclear membrane in cardiac and skeletal muscle [39,40] that interacts with the N-terminal domain of LAP1 [38]. Loss-of-function mutations in *EMD*, the gene that encodes emerin, cause X-linked Emery–Dreifuss muscular dystrophy and cardiomyopathy in humans [41]. The interaction of emerin with LAP1 was first reported by Shin et al. in 2013, who showed that combined deletion of the genes encoding LAP1 and emerin led to worsened phenotypes of muscular dystrophy and shortened lifespan in mice when compared to *LAP1*- or *EMD*-null mice [38]. Interestingly, *EMD*-null mice did not develop muscle pathology [38]. Of note, previous reports have documented the absence of muscle involvement in other mouse models lacking emerin [42,43]. The level of LAP1 in humans is about half of that in mice; conversely, emerin levels are significantly higher in humans compare to mice [38]. It has therefore been postulated that mouse skeletal muscle cells may be less dependent on emerin than human cells, and that high levels LAP1 compensate for low levels of emerin [38]. Overall, evidence indicates that the interplay between LAP1 and emerin is crucial for maintenance of normal striated muscle integrity and function. Further examination showed that, in LAP1-depleted fibroblasts, emerin and A-type lamins were abnormally localized along the nuclear envelope, whereas other proteins in the nuclear envelope were distributed normally [38]. This implies that LAP1 is necessary to ensure laminA and emerin proper position at the inner nuclear membrane.

#### 3.2.3. TorsinA

TorsinA is an AAA+ protein encoded encoded by the TOR1A gene (OMIM #605204). This member of the ATPase family is preferentially expressed in neural tissues and acts as a molecular chaperone [44,45]. TorsinA resides within the endoplasmic reticulum and perinuclear space and interacts with the C-terminal domain of LAP1 [17,46,47,48]. Of note, LAP1 also binds to another torsin family member, torsinB [48,49]. LAP1 recruits torsinA to the nuclear envelope and modulates TorsinA ATPase activity through the use of a arginine finger located in the LAP1 luminal domain [17,49,50,51]. LULL-1 is another torsinA activator, which has a C-terminal domain highly similar to that of to LAP1 [17]. Reciprocally, torsinA is thought to act as a modulator of the interaction of LAP1 with chromatin [52].

The physiological relevance of the functional interplay between LAP1 and torsinA was demonstrated by Kim et al. in 2010, who showed that *TOR1AIP1*-knockout mice have morphological abnormalities in their neuronal nuclear membranes (i.e., blebbing) [27], similar to those observed in neurons from DYT1 mice model (Tor1aΔE/ΔE) lacking torsinA [17,48]. Moreover, in fibroblasts in the absence of LAP1, the localization of torsinA shifted from the nuclear envelope toward the endoplasmic reticulum [48]. Hence, in fibroblasts, LAP1 is responsible for the localization of torsinA to the inner nuclear membrane. Although LAP1–torsinA interactions impact multiple tissues, including the heart, liver, kidney, and skin, neurons are most susceptible to changes in the LAP1–torsinA interactions [48]. This susceptibility seems to stem from the involvement of additional proteins (i.e., torsinB, printor, nesprin-3α) that interact with the torsinA–LAP1 pathway [48]. A review of the LAP1 interactome supported the importance of the LAP1–torsinA interaction, ranking “chaperone-mediated protein folding requiring cofactor” as one of the most relevant biological processes attributed to LAP1 interactors in humans [27].

In humans, a glutamic acid deletion in torsinA caused by pathogenic variants in the *TOR1A* gene results in early-onset primary dystonia, a hyperkinetic movement disorder [53,54]. This mutation can alter the binding of torsinA to LAP1, but the exact role of LAP1 in dystonia remains unknown [46,49,55]. Interestingly, one patient carrying a *TOR1AIP1* mutation affecting the torsinA binding domain of LAP1 presented with dystonia, dilated cardiomyopathy, and cerebellar atrophy [3]. Although the patient’s muscle biopsy did not reveal any structural abnormalities, levels of LAP1 were abnormally low, and LAP1 was mislocated in the endoplasmic reticulum [3]. The LAP1–torsinA interaction has also been linked to cell migration in humans [56] (as discussed in more detail in Section 4.2). Furthermore, the LAP1–torsinA interaction may be important for lipid homeostasis, as conditional knockdown of LAP1 in hepatocytes in mice results in mild steatosis and accumulation of nuclear lipid droplets [57,58].

#### 3.2.4. PP1 and TRF2

The protein phosphatase PP1 is a ubiquitous regulator of cellular functions ranging from glycogen metabolism to transcription [13]. PP1 is located in the nucleus and has been shown to interact with the N-terminal domain of LAP1B in vitro and in vivo [18,59]. PP1 dephosphorylates two of the five phosphorylated residues of LAP1B, Ser306 and Ser310 [13]. TRF2, or telomeric repeat-binding factor 2, is involved in the DNA damage response (DDR) processes that repair double-strand breaks occurring at extratelomeric chromatin regions [60]. TRF2 was shown to interact with LAP1 in human cells lines in response to DNA damage induced by PP1 and PP2A-inhibiting agents [60]. Their findings showed that, when phosphorylated, LAP1 has a greater affinity to TRF2 [60]. The same authors postulated that PP1-mediated dephosphorylation could favor TRF2–LAP1 complex separation.

## 4. The Biological Processes Involving LAP1 in Humans

The physiological roles of LAP1 in humans have not been entirely deciphered; however, evidence from animal models, studies of cultured human cells, and bioinformatic analyses have linked LAP1 to a large variety of cellular pathways and biological processes. Moreover, the clinical relevance of LAP1 in human physiology is supported by the absence of homozygous deletion, duplication, or nonsense mutations in *TOR1AIP1* in the healthy population [5] and the association of mutations in *TOR1AIP1* with tissue-specific pathological processes [3,4,5,6,7,8,10,12] (OMIM #614512). LAP1 has highly specific roles in restricted cell or tissue types and ubiquitous functions in DNA repair and mitosis as described in this section. Although not discussed in detail here as experimental validation is lacking, bioinformatic gene-hub studies have linked LAP1 to various biological processes and diseases including carcinogenesis, dysferlinopathy, and glucocorticoid treatment response in patients with vitiligo [61,62,63]. Of note, evidence indicates that LAP1 is involved in the secretion of very-low-density lipoprotein in the liver of mice [57,58], but not yet in humans.

### 4.1. Nuclear Envelope Structure and Chromatin Positioning

As described above, LAP1 is important for nuclear envelope integrity and chromatin positioning through its interaction with lamins A/C and B, as demonstrated in animal models [15,36,38]. Studies investigating protein–protein interactions in humans using yeast two-hybrid screening, as well as mass spectroscopy, identified LAP1 as a member of the lamin A [64,65] and lamin C [65] interactomes. In line with this results, Gene Ontology enrichment term analysis and Ingenuity pathway analysis strongly implicated LAP1 in the maintenance of nuclear organization and structure in human cells [27].

### 4.2. Mitosis and Cellular Dynamics

Extensive evidence indicates that LAP1 is needed for somatic cell division, a process consisting in the breakdown and subsequent reformation of the nuclear envelope to allow chromosomes to reach the mitotic spindle. The latter is a dynamic structure, constructed from microtubules, that allows the segregation of chromosomes during cell division [66,67]. Notably, LAP1 seem to separate from lamins and chromosomes during early mitosis, thus contributing to NE remodeling [68]. In HeLa cells depleted of LAP1, mitotic spindle formation during prometaphase is impaired resulting in cellular death, functionally linking LAP1 with mitotic processes [69]. The colocalization of LAP1 with acetylated α-tubulin in the mitotic spindle and with α-tubulin in the centrosomes of mitotic cells has been demonstrated [70]. Furthermore, centrosomes are abnormally positioned, and fewer microtubule stability markers are expressed in cells that are deficient in LAP1 [70]. Proteomic studies have linked AP1 to MAD2L1 and MAD2L1BP in the mitotic checkpoint complex [71,72], which is an indirect regulator of the chromosome segregation process. LAP1 is also implicated in nuclear envelope reassembly in rat cells [16]. Lastly, LAP1 levels vary during the cell cycle and are highly phosphorylated during mitosis in human cells in culture, suggesting that its interaction with PP1 might be involved in cell-cycle progression [70].

### 4.3. Cell Migration

During fibroblast migration, the centrosome is relocated to the leading edge of the cell, and the nucleus is driven backward by perinuclear actin filaments attached to transmembrane actin-associated nuclear lines [73,74,75]. In fibroblasts, both transmembrane actin-associated nuclear line assembly and retrograde movement of actin cables require LAP1 and torsinA [56]. As these nuclear movement are necessary to reorient the centrosome, it follows that LAP1 contributes to appropriate positioning of migrating centrosomes in fibroblasts [56]. *TOR1AIP1* mutations that cause loss of LAP1B and LAP1C were identified in seven patients with severe multisystemic disorders. Functional studies demonstrated deficient migratory capacity in fibroblasts from two of these patients [10].

### 4.4. DNA Damage Response and Genome Integrity

The DNA damage response is mediated by protein kinases including ATM and ATR [76,77,78], and LAP1 is a substrate of ATM as demonstrated in large-scale proteomic analyses [76]. Additional proteomics studies have linked LAP1 to other kinases including EGFR [79], TrkA [80], and LRRK2 [81] that are important for genome integrity. Other actors in the response to DNA damage identified in LAP1 interactome include TRF2 [60], TERF2IP [82], and RIF1 [83], which are known to be part of the shelterin complex, a protein structure involved in chromosome end protection and telomere length regulation [84]. Disruption of this complex activates ATM [85]. Pereira et al. showed recently that the LAP1–TRF2 complex colocalizes with other DNA damage response proteins in response to H_2_O_2_- and bleomycin-induced DNA damage in HeLa cells [60]. Such evidence support the idea that LAP1 plays a role in DNA damage response. LAP1 implication in this process was further supported by informatic studies [27].

### 4.5. Spermatogenesis

It is well recognized that the nuclear envelope undergoes structural transformation during spermatogenesis [86,87]. Although roles of other nuclear proteins in spermatogenesis have been previously established, the first evidence implicating LAP1 in spermatogenesis-specific nuclear envelope remodeling came in 2017 when Serrano et al. reported dynamic changes in the location of LAP1 during spermatogenesis [86]. LAP1 first localizes on the centriolar pole and then relocates to the posterior pole along with chromatin during spermatid maturation. LAP1 is thought to facilitate the systematic repositioning of chromatin, as well as contributing to the development of manchette [86].

### 4.6. Skeletal Muscle Maintenance and Growth

Evidence from experiments in mice suggest that emerin and LAP1 function together to preserve skeletal muscle function and integrity [38]. Depletion of LAP1 in mouse models did not substantially alter early myogenesis, although it clearly impacted the late embryonic and postnatal muscle growth [38]. It has been suggested that LAP1 is necessary for hypertrophic muscle growth and satellite cell reactivation [19]. During differentiation of mouse myoblasts, levels of in LAP1A and LAP1B increase, but LAP1C levels remain constant, potentially indicative of LAP1A- and LAP1B-specific properties during myogenesis [19]. *TOR1AIP1* mutations have been reported in patients with skeletal muscle disorders with or without cardiomyopathies. In muscle biopsies from several of these patients, central nuclei, myofiber degeneration, and/or ultrastructural alterations of the nuclear envelope were observed [4,5,6,7,9,11,12], suggesting that LAP1 is necessary for the integrity of myonuclei and normal muscular structure in humans.

### 4.7. Cardiac Muscle Function

That LAP1 is important in cardiac muscle function is clinically suggested by the frequent cardiac involvement in patients with mutations in *TOR1AIP1* [3,4,8,9,10,12]. In mouse models, selective depletion of LAP1 in cardiomyocytes resulted in left-ventricular dysfunction [88]. However, no overt arrythmia was noticed, a feature which is associated with *TOR1AIP1* mutations in humans [8,9]. There were no major morphological changes in cardiac muscle in mice deficient in LAP1 [20]; the only heart biopsy of a human with a *TOR1AIP1* mutation reported to date showed disrupted myocytes with fibrotic and adipose infiltrates but no ultrastructural or nuclear abnormalities [8].

### 4.8. Neuromuscular Transmission

The link between LAP1 and neuromuscular transmission is suggested by the congenital myasthenic syndrome (CMS) phenotype that was first described in two patients with *TOR1AIP1* mutations. Muscle biopsies of these patients revealed nuclear abnormalities reminiscent of those observed in patients with other *TOR1AIP1*-related disorders [7]. Three other patients were subsequently identified as having *TOR1AIP1* pathogenic variants causing CMS [4,89]. Mice deficient in LAP1 develop progressive muscular weakness and susceptibility to fatigue coinciding with compound motor action potential decrement [7], a typical electrophysiological feature of myasthenic syndrome. Skeletal muscle biopsies M-LAP1-/- mouse with CMS phenotype revealed morphological anomalies of the neuromuscular junctions including a high number of synaptic myonuclear and an abnormal expression of neuromuscular junction proteins [7]. Therefore, the authors suggested that the absence of LAP1 could potentially affect neuromuscular transmission by disrupting neuromuscular junction structure. However, the possibility that any observed effects are caused by other factors related to muscle regeneration and generation processes could not be ruled out [7].

## 5. Mutations in the *TOR1AIP1* and Associated Phenotypes

Kayman-Kurekci et al. were the first to report an LAP1-associated phenotype in humans; in 2014, these authors identified three affected patients from a Turkish family who each carried a frameshift mutation (c.186delG) in the *TOR1AIP1* gene. To date, only 23 patients have been identified with mutations in *TOR1AIP1*, and all pathogenic variants were inherited in a recessive manner [3,4,5,6,7,8,9,10,11,12]. Similarly to what has been described in other nuclear envelopathies, the clinical spectrum reported in the literature is broad, ranging from isolated tissue-specific disease to complex multisystemic disorders. Patients are of both genders and various ethnicities. Consanguinity has frequently been highlighted in the familial history as in many rare, recessively inherited conditions. The age of symptom onset is variable, from in utero stage to early adulthood. The level of evidence for causative roles of the *TOR1AIP1* mutations varies greatly across studies (Table 1). The cases reported to date are summarized in Table 1 and discussed in the sections below.

### 5.1. Cardiomyopathies and Congenital Heart Defects

Cardiomyopathies have been reported in seven patients, of which three displayed multisystemic clinical presentation [3,9], one had isolated cardiomyopathy [8], and three had muscular dystrophy [5,8,12]. The patient with isolated cardiomyopathy was an 11 year old boy in which potential muscular involvement at a later stage cannot be excluded [8]. Two of these patients were reported to have self-resolved ventricular septal defect at birth [9]. Cardiac involvement included dilated cardiomyopathy, left-ventricular dysfunction, left-ventricular hypertrophy with preserved ventricular systolic function, and arrhythmia. The severity of cardiac involvement ranged from mild concentric left-ventricular hypertrophy to severe end-stage chronic heart failure [3,5,8,9,12]. Patients were treated with various heart medications; two required a heart transplant in their second decade [8]; and one experienced acute heart failure leading to kidney failure and chronic dialysis [9].

In addition to the two patients with transient ventricular septal defect mentioned before, [9], six subjects presenting with congenital heart defects and a multisystemic syndrome with progeroid features were reported [9,10]. Three of them required invasive surgery, and four died in their first decade, although the exact cause of death was not reported [10].

Cardiac involvement seems to be a prominent characteristic of *TOR1AIP1* mutations and can be life-threatening, underscoring the importance for close monitoring.

### 5.2. Muscular Dystrophies and Myopathies

*TOR1AIP1* mutations cause a recessive limb-girdle muscular dystrophy, which was formerly referred to as LGMD2Y [90]. The first patient reported displayed proximo-distal weakness with contractures, mild cardiopathy, and restrictive lung disease [12]. Since then, muscular dystrophy has been described in six other patients with *TOR1AIP1* mutations [5,6,8,11] who do not necessarily share all features reported in the original patient. The distribution of muscle weakness and wasting varies and is often combined proximal and distal involvement and more rarely axial weakness [5,6,8,11,12]. Contractures are a common occurrence, primarily affecting the distal and cervical regions, which may result in a rigid spine. Ptosis was reported in one patient, although there was no report of nerve conduction studies, which would have ruled out a concomitant dysfunction of neuromuscular transmission [9]. Creatine kinase levels are usually normal to mildly elevated. One patient with an unspecified compound heterozygous *TOR1AIP1* variant was reported to have highly elevated creatine kinase (10,000 U/L). The muscle biopsy was remarkable for inflammation (upregulation of MCH-I and sarcolemma staining for complement), necrosis, and regeneration. Muscle magnetic resonance imaging showed fatty degeneration of the proximal muscle of lower limbs and abnormally high T2 signals in her lower legs. Electromyography was compatible with an irritable myopathy. None of these features were reported in any other patient, and we could not find the confirmation in the article that autoimmune inflammatory myopathies were ruled out [11]. Cardiomyopathy was reported in three of these seven subjects, as mentioned in the previous section [6,8,12]. Respiratory involvement characterized by restrictive syndrome was clinically documented in three patients [5,12].

### 5.3. Congenital Myasthenic Syndrome

CMS is a rare genetic condition characterized by impaired function of the neuromuscular junction, leading to muscle weakness that worsens with physical exertion. In 2020, two siblings were reported with proximal muscular weakness and clear electrophysiological evidence of neuromuscular junction impairment [7]. Further investigation revealed that the two shared a homozygous frameshift mutation p. (Pro43fs*15) in *TOR1AIP1*, which resulted in the absence of LAP1B isoforms in cultured muscle cells [7]. In 2021, three additional patients with *TOR1AIP1*-induced CMS were identified in a consanguineous Algerian family. Symptom onset varied from early infancy to 20 years of age. The three affected siblings shared common dysmorphic features, prominently proximal weakness with various degrees of distal and axial weakness, distal muscle wasting, and muscle fatigability. Mild cervical spine rigidity was reported in two of the three siblings [4]. Ptosis, ophtalmoparesis, and facial and bulbar weakness, which are common findings in CMS [91], were not reported. Creatine kinase levels were normal to slightly elevated (maximum 401 U/L), and none of the patients exhibited cardiac or respiratory involvement. Treatment with acetylcholine inhibitor pyridostigmine alleviated muscle weakness and fatigability in two of the five subjects and improved exercise tolerance in another [4,7], stressing the importance of accurate diagnosis of this rare treatable condition.

### 5.4. Multisystemic Disorders

#### 5.4.1. Multisystemic Disorders with Progeroid Features

Combinations of congenital-onset multisystemic symptoms and progressive progeroid syndromes have been reported in nine patients with *TOR1AIP1* mutations from three distinct families of European and Palestinian descent. Six of these subjects came from consanguineous families or areas with high inbreeding rates. Symptoms were first noticed in utero or at birth [10]. Progeroid syndromes are a heterogeneous group of diseases mainly characterized by signs of premature aging leading to early death [92]. Features reported in these subjects with *TOR1AIP1* mutations previously observed in patients with these syndromes include hearing loss, bilateral early-onset cataract, skin abnormalities, cranial deformities, facial dysmorphism, and growth retardation. Neurodevelopmental delays of varying severities have been reported in the subjects with *TOR1AIP1* mutations. Neurological abnormalities including microcephalia, as well as cortical and ponto-cerebellar atrophy, were seen in the most affected patients [10]. Neuromuscular symptoms include proximal weakness [9] and generalized hypotonia [10]. Cardiac involvement was reported in all but one patient; symptoms reported were dilated cardiomyopathy, arrhythmia, and congenital heart defects [9,10]. Dystonic posture with choreoathetosis movements was reported in one patient [9]. Consistent with other progeroid syndromes, life expectancy was reduced, as four patients died within their first decade, along with one before 20 years of age.

#### 5.4.2. Other Multisystemic Presentations

A unique presentation combining severe dystonia, cardiomyopathy, and cerebellar atrophy was described in a single Moroccan patient [3]. Of interest, the dystonia was reported to be extremely painful and resistant to most pain medications but was eventually improved by treatment with nabiximols. The subject died at age 17 due to dilated cardiomyopathy [3].

### 5.5. Pathology and Imaging in TOR1AIP1-Related Disorders

Muscle biopsies were performed in patients with different clinical presentations at various ages and disease stages, likely accounting for the large variability in the findings. Overall, when present, findings in muscle optic microscopy were nonspecific, including mild myogenic features, inflammatory [11] or dystrophic pattern [6,8]. Sarcomere structures observed by electronic microscopy were usually normal [4,6,8,12,93], although ultrastructural changes were reported in one patient [5]. Evidence of nuclear and nuclear envelope disruption was common [5,6,7,12,93]. Immunofluorescence of patients’ fibroblasts provided further evidence for nuclear involvement such as nuclear blebs, nuclear lobulation and nuclear envelope invagination. These characteristics are reminiscent of those of cells from Tor1a mutant animal models and of samples from human subjects with progeroid syndromes [48,92]. Cytoplasmic channels were described in fibroblasts from some subjects [9,10]. Expression of other lamins and emerin was sometimes abnormal, but inconsistent results allowed no conclusion to be drawn [9,10].

Although skeletal muscle involvement was clearly demonstrated by muscle MRI of five patients, there were no specific patterns. However, knowing that muscular involvement in these conditions can be identified through imaging can be relevant to guide muscle biopsy and allow further functional studies based on the affected muscle sample.

### 5.6. Evidence for Phenotype–Genotype Correlations

We already discussed some evidence emphasizing the importance of LAP1B in myogenesis. Homozygous mutations p.(Arg22Glnfs*88), p.(Gln35fs*74), and p.(Pro43fs*15) are all located in exon 1, between the two alternative *TOR1AIP1* start codons. These mutations are predicted to truncate LAP1B only, as confirmed by several analyses of mRNA and protein in patient samples [5]. All patients carrying these homozygous mutations presented with muscular phenotypes (i.e., myopathy or CMS) and similar pathological features. As a result, it has been proposed that mutations affecting LAP1B alone, while leaving LAP1C unaffected, could be responsible for skeletal muscle phenotype [5]. In some of these patients, LAP1C was upregulated, suggesting that LAP1C cannot fully compensate for LAP1B deficiency or absence [12]. Of note, some mutations have been associated to distinct phenotypes. The homozygous mutation p.(Arg22Glnfs*88) was linked to both CMS [4] and myopathy [5]. However, nerve conduction studies of the patient presenting with the latter phenotype demonstrated abnormally low compound motor action potential, but whether this is indicative of CMS is unclear [5]. The mutation p.(Pro43fs*15) was first identified in the heterozygous compound state in association with p.(Leu394Pro) in two patients with severe dilated cardiomyopathy with or without muscular dystrophy [8]. Neither underwent electrophysiology. Whether the compound mutation p.(Leu394Pro), which is predicted to affect both isoforms of LAP1, is actually responsible for the more severe cardiac involvement remains unclear. Homozygous mutation p.(Pro43fs*15) was reported in two patients diagnosed with CMS without cardiac involvement [7]. On the other hand, mutations predicted to affect both LAP1 isoforms are mainly linked to multisystemic presentation with or without progeroid features. As LAP1C levels are higher in undifferentiated cells than differentiated cells in animal models [19,20], this isoform may be important in the early development of the brain and other tissues. Interestingly, patients with progeroid features tend to have homozygous variants truncating both isoforms [10] or heterozygous compound variants with at least one truncating variant [9] that result in complete absence of both isoforms in patients’ fibroblasts [9,10]. By contrast, cardiomyopathy, cerebellar atrophy, and dystonia was observed in one patient with homozygous missense variants that result in reduced levels (not absence) of both isoforms. Whether residual isoform expression modulates the overall phenotype of *TOR1AIP1* mutations is unclear. Residual LAP1C could mitigate the phenotype as the most severely affected patients have complete loss of both isoforms [10]. Some proposed that brain involvement is linked to abnormally low levels of LAP1C and that residual LAP1B could account for the normal muscular biopsy found in patient harboring the p.(Glu482Ala) mutation [5]. The same authors postulated that the heterozygous missense mutation p.(Leu349Pro), expected to affect exon 10 and, thus, both isoforms, could be responsible for the more severe cardiac involvement seen in the two siblings carrying compound heterozygous mutations [5,8]. However, LAP1C expression was not evaluated in these patients [8]. The p.(Glu482Ala) mutation results in the substitution of a highly conserved glutamic acid by alanine in both isoforms; this amino acid is located in the C-terminal domain, the region necessary for interaction with torsinA [17]. Interestingly, dystonic posture and choreoathetosis were also reported in the patient heterozygous for p.(Glu217*) and p.(Asp242fs*17) [9], but not in the patient carrying compound heterozygous p.(Pro43fs*15) and p.(Leu394Pro) mutations [8], although the latter is located within the torsinA-interacting domain. Although there are some hints indicating a genetic–phenotypic correlation, some limitations must be kept in mind. First, whether loss or deficiency in one or both LAP1 isoforms determines the cardiac and neuromuscular junction involvement, as well as the severity of symptoms, remains unclear. Secondly, examination was incomplete with limited follow-up and varying types of investigations in many subjects, which may have biased one’s interpretation. Thirdly, additional mutations or variants that have not yet been identified may impact phenotype. Lastly, one must bear in mind the different level of evidence used to assess whether the TOR1AIP1 is truly responsible for the phenotype.

## 6. Conclusions

Although strong evidence links LAP1 to numerous biological processes, the exact mechanisms underlying its functions and relevance to human diseases are not yet entirely deciphered. Most evidence of the importance of LAP1 in human physiology and disease is based on in vitro and in vivo studies in animal models. Limitations of these models include the presence of an additional isoform (LAP1A) in rodents and substantially higher levels of LAP1 when compared to human [38]. Moreover, the tissue-specific distribution and ratio of LAP1 isoforms varies across species, suggesting that their roles might not necessarily be identical. Therefore, further evidence will be needed in order to elucidate the precise role of and mechanisms of LAP1 in humans. With respect to the LAP1 complex interactome and multiple functions, proteomics is a promising approach to better delineate the biological properties of the LAP1 isoforms.

Reflecting the complexity of LAP1 interactions, mutations in the *TOR1AIP1* gene have been reported in association with a heterogenous phenotypic spectrum, similar to those of other nuclear envelopathies. Evidence collected to date suggests an association between the location and the type of mutation in *TOR1AIP1* with clinical presentation, which ranges from isolated cardiomyopathy to fatal progeroid-like multisystemic disorders. Refining clinical examination and phenotyping will help to better delineate the clinical spectrum linked to *TOR1AIP1* mutations. Lastly, population, segregation, and functional studies will be needed to evaluate the pathogenicity of new and previously described variants.

With regard to therapeutics, three patients with *TOR1AIP1*-related CMS benefited from treatment with pyridostigmine [4,7], which stresses the need for early identification of these patients with neuromuscular transmission dysfunction. Strict cardiac follow-up is necessary for all patients with disease-causing *TOR1AIP1* mutations to allow timely medical intervention.

From an experimental point of view, several disease-causing mutations in *TOR1AIP1* described to date result in premature stop codons that truncate both isoforms or only LAP1B. Therefore, read-through and exon-skipping therapies, similar to those used in the treatment of other rare neuromuscular diseases [94], could be investigated. It would also be interesting to assess whether strengthening LAP1 interactions with various cofactors (e.g., emerin and torsinA) could modulate the phenotypes of affected subjects. Additionally, as Shin et al. showed that increased LAP1 levels correlated with milder phenotypes in *emerin*-null mice [38], LAP1 could reciprocally modulate other diseases phenotypes. Compensatory approaches that involve upregulating the expression of surrogate proteins to compensate for the absence of a specific protein have been proposed notably in Duchenne muscular dystrophy [95] and several congenital myopathy [96]. Exploring the potential of LAP1 overexpression to alleviate other disease phenotypes is a topic that warrants further investigation. Lastly, further work is warranted to determine how human LAP1 isoforms function, to refine our understanding of the LAP1 interactomes, and to delineate phenotypes associated with pathogenic mutations.

## Figures and Tables

**Figure 1 ijms-24-06911-f001:**
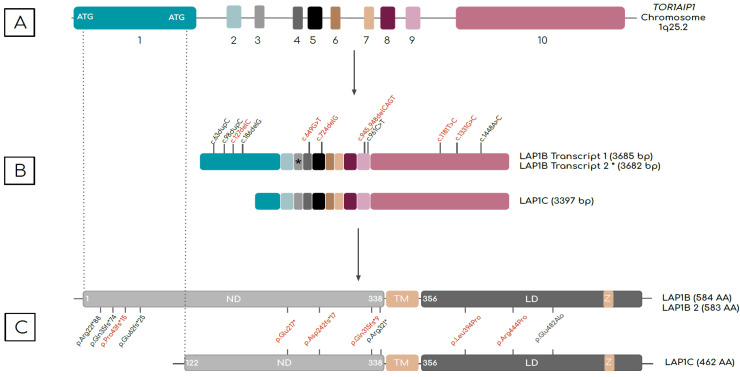
Schematic of (top to bottom) human *TOR1AIP1* gene, the transcript variants, and LAP1 protein isoforms. (**A**) *TOR1AIP1* has 10 exons represented by colored boxes. Two start codons are found in exon 1. Alternative translation start sites produce distinct transcripts. (**B**) Two LAP1B transcripts have identical sequence apart from a CAG trinucleotide in exon 3, indicated by an asterisk (*). (**C**) LAP1C is truncated at the N-terminus relative to LAP1B. LAP1B and LAP1C proteins, which differ in the number of amino acids (AA) at the N-terminus, have three distinct domains: an N-terminal nucleoplasmic domain (ND), a transmembrane domain (TM), and a C-terminal luminal domain (LD). A zinc-finger motif (Z) is present in the C-terminal regions of both proteins. Mutations reported to date are indicated in transcript and protein schematics. Compound heterozygous mutations are indicated in red. Mutations that affect LAP1B seem to be associated with muscle involvement, whereas mutations that affect both isoforms result in early-onset multisystemic presentation.

**Figure 2 ijms-24-06911-f002:**
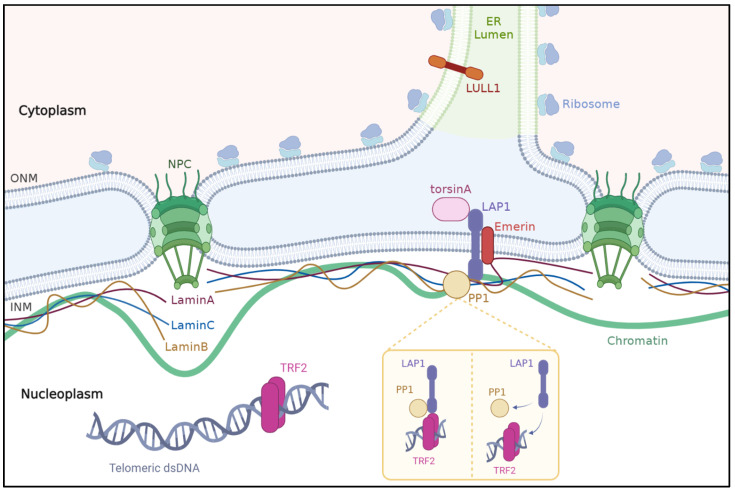
Schematic of LAP1’s main interactions. The nuclear envelope formed by the outer and inner membranes (ONM and INM, respectively). The latter is underlaid by a network of type V intermediate filaments (A/C- and B-type lamins) forming the basal lamina. ONM is in direct contact with the endoplasmic reticulum. NE contains transport channels called nuclear pore complexes. LAP1 is located on the INM and interacts with emerin and lamins, either directly or via emerin, contributing to maintaining skeletal muscle and NE structure integrity. Activation of TorsinA ATPase activity requires interaction with LAP1 at the NE or LULL1 at the ER. PP1 mediates LAP1’s Ser306 and Ser310 residues dephosphorylation. In response to DNA damage, phosphorylated LAP1 interacts with TRF2 that is part of the shelterin complex, a structure ensuring telomer repair and genome integrity. ER: endoplasmic reticulum; INM: inner nuclear membrane; LAP1: lamina-associated protein 1; LULL1: luminal domain-like LAP1; NE: nuclear envelope; NPC: nuclear pore complex; ONM: outer nuclear membrane; PP1: protein phosphatase 1.

**Table 1 ijms-24-06911-t001:** Summary of *TOR1AIP1* mutations and level of evidence.

Mutation	Phenotype	Mutation Type	Isoform Affected	Genetic Testing	Segregation Study	Population Database	DNA and RNA Analyses	Protein Analysis ^1^	Immunofluorescence	Material	Other Evidence	Ref
c.63dupC p.(Arg22Glnfs*88)	CMS	Frameshift (premature stop codon and loss of function)	LAP1B	Panel	Yes	NR	NR	NR	Signal from LAP1 antibody (both isoforms), nearly absent signal in myocytes	Muscle biopsy	Skeletal muscle OM, EM; muscle MRI; ENMG	[4]
c.63dupC p.(Arg22Glnfs*88)	MD	Frameshift (premature stop codon and loss of function)	LAP1B	WGS	No	Yes	Reduction of 16% in mRNAs encoding LAP1B + LAP1C; *TOR1AIP1* cDNA levels at 6% of control’s	Strong reduction of LAP1B levels; normal LAP1C levels	NR	Fibroblasts	Skeletal muscle OM, EM; muscle MRI	[5]
c.98dupC p.(Gln35fs*74)	MD	Frameshift (premature stop codon)	LAP1B	WGS	Yes	Yes	NR	NR	NR	NA	Skeletal muscle OM, EM; muscle MRI	[6]
c.127delC p. (Pro43fs*15)	CMS	Frameshift (premature stop codon)	LAP1B	WES	Yes	NR	NR	No LAB1B isoform; normal expression of LAP1C	No expression of either LAP1 isoform in myonuclei	Muscle biopsy; cultured muscle cells	Animal model; skeletal muscle OM, EM	[7]
**c.127delC** **p.(Pro43fs*15)**	MD and/or dilated cardiomyopathy	Frameshift (premature stop codon)	LAP1B	WES	Yes	Yes	NR	No LAP1B isoform; normal levels of Lamin A/C and emerin.	Normal to slightly elevated Lamin A/C, LAP2B, emerin	Muscle biopsy; cardiac biopsy	Skeletal muscle OM, EM; cardiac muscle OM, EM; muscle MRI	[8]
**c.1181 T > C p.(Leu394Pro)**		Missense (change in sequence in exon 10)	LAP1B + LAP1C									
c.186delG p.(Glu62fs*25)	MD	Frameshift (premature stop codon)	LAP1B	WGS	Yes	Yes	Fivefold decrease in *TOR1AIP1* mRNA levels compared to control’s	No LAP1B; Increased expression of 50-kDA isoform (LAP1C?); increased expression of LULL1.	No LAP1B staining in Myonuclei and faint staining in endomysial cell nuclei; normal staining for LULL1, torsinA, and lamin B	Muscle biopsy	Skeletal muscle OM, EM	[12]
**c.649G > T** **p.(Glu217*)**	MS (progeroid)	Nonsense	LAP1B + LAP1C	WES	Yes	Yes	NR	NR	NR	NA	Skeletal muscle OM; brain MRI	[9]
**c.724delG p.(Asp242fs*17)**		Frameshift (premature stop codon)	LAP1B + LAP1C									
**c.945_** **948delCAGT** **p.(Gln315fs*9)**	MS (progeroid)	Frameshift (premature stop codon)	LAP1B + LAP1C	WES	Yes	Yes	No difference in *TOR1AIP1* mRNA levels between patients and controls; reductions in *LMNA* and *LMNB1* mRNA levels	Absence of LAP1B and LAP1C; reduction in lamin B1 (senescence marker) and lamin A/C levels	Loss of lamin A/C staining in 18% of analyzed nuclei; lamin A/C aggregates at the nuclear periphery and nucleoplasm; NE invagination (~90% of abnormalities highlighted); complex nuclear lobulation; nuclear blebs; nuclear holes (4%) resembling cytoplasmic channels	SF	Brain CT; ENMG	[9]
**c.1331G >C** **p.(Arg444Pro)**		Missense (affects conserved arginine residue)	LAP1B + LAP1C									
c.961C > T p.(Arg321*)	MS (progeroid)	Nonsense (premature stop codon)	LAP1B + LAP1C	WES	Yes	Yes	Decreased levels of mRNA encoding LAP1 and slightly elevated emerin mRNA	Absence of LAPB1B and LAP1C	Reduced intensity of lamin A/C nuclear rim staining; distorted nuclei; cytoplasmic channels in fibroblasts nuclei (6.3%)	SF	Brain MRI	[10]
c.1448A > C p.(Glu482Ala)	MS	Missense (affects highly conserved glutamic acid)	LAP1B + LAP1C	WGS	Yes	Yes	NR	Reduced levels of LAP1 isoforms	Reduction in LAP1 staining in the NE; mislocation and aggregation of LAP1 in the ER	SF	Bioinformatic prediction	[3]
Unspecified	MD	NR	NR	WGS	NR	NR	NR	NR	NR	NR	Skeletal muscle OM, EM; muscle MRI	[11]

^1^ When analyzed, proteins were identified by Western blot. Compound heterozygous mutations are in bold. Abbreviations: CMS, congenital myasthenic syndrome; EM, electron microscopy; ER, endoplasmic reticulum; MD, muscular dystrophy; MRI, magnetic resonance imaging; MS, multisystemic; NA, not applicable; NCS/EMG, electromyography and nerve conduction studies; NE, nuclear envelope; NR, not reported; OM, optical microscopy; SF, Skin Fibroblasts; WEG, whole-exome sequencing; WGS, whole-genome sequencing.

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
