# Peer review of "TOR1AIP1*-Associated Nuclear Envelopathies"

_ijms, 2023, doi:10.3390/ijms24086911_

Round 1
Reviewer 1 Report
This is a well-written review of our current knowledge of the TOR1AIP1 gene, some minor points the authors may wish to consider:
1. I found the constant switching between gene name and a totally different protein name somewhat distracting. Whilst I realise just using one may be impossible, particularly when describing isoforms, would it be possible to focus more consistently on one name or another?
2. I would not describe Ref. 28 as a bioinformatic study - its a literature review. Also now somewhat dated - have the authors checked in interaction databases such as IntAct (www.ebi.ac.uk/intact) to see if any additional information is available?
Reviewer 2 Report
The review entitled: “TOR1AIP1-Associated Nuclear Envelopathies” addresses the role of TOR1AIP1 in human pathology. Overall, the authors do a comprehensive reviewing of the underlying literature. The authors explain very well the different LAP1 isoforms and provide a very informative list of LAP1 associated diseases (Table 1). Key facts about LAP1 function are provided, however, what is missing in the review is a section that captures (discusses) the molecular mechanisms that underpin TOR1AIP1-Associated Nuclear Envelopathies. Which LAP1 biological functions (cell polarity; DNA damage; nuclear architecture,etc) are key (stratification) for which tissues and/or organs (material for an additional figure) ? As things stand the review reads as a list of facts without providing a digest of LAP1 current functions and how this manifest human disease. The above is important as it will enhance our understanding of why TOR1AIP1 disfunction yields disease and further increase the quality of the review.
Minor omissions and matters that require further attention:
Key word: inner nuclear envelope may be better if it is substituted with inner nuclear membrane?
Some abbreviations have not been properly introduced (e.g. INM; ONM, NE, CMS, etc)
Line 58-59; and line 107: The generation of TOR1AIP1 requires more clarity. The authors suggest a number of different mechanisms but then state that: “The two protein isoforms result from 60 different translational initiation sites”.
Line 66-67: In what context is LAP1B more soluble (e.g. in vivo, in vitro studies; specifying the term “soluble” will be beneficial)?
Line 70: Correct “type 2” to type II
Lines 86-87: Correct sentence to: “Specifically, higher levels of LAP1C are detected in undifferentiated human cell 86 lines”; “whereas LAP1B is expressed at higher levels in highly specified tissues such as brain 87 and heart [13,21]”. What do the authors mean with the word “specified”?
Line 93: Full stop in the sentence is missing.
Line 101: Correct the sentence: “there is some level of experimental evidence for 36 of these interactions in hu-101 mans [28].”
Line 102: Specify the type of lamins and torsins that interact with LAP1.
Lines 113-114: I cannot detect anything labelled in orange.
Line 126: The diseases that are linked to LMNB1 are not up to date.
Lines 138-140: Clarity and more detail is required with respect to the statement: “which had been previously observed by other groups”.
Line 157: “a structure ensuring telomers repair and 157 integrity.”
Line 164: TosinA
Line 190: remains unknow
Line 196: lipidic
Line 239: correct the name of “-tubulin”
Line 249: Define the type of cell in sentence “During cell migration, the centrosome is relocated to the leading edge of the cell…); In T-cells for example the mechanism of movement is different.
Line 253: “During fibroblast migration, LAP1 contribute to appropriate positioning of 253 migrating centrosomes [55].” This sentence contradicts previous statements that mentioned nuclear migration rather than centrosome migration.
Line 265: Involved
Line 329: thus later muscular involvement cannot be excluded
Line 338: six amore
Line 356: Her muscle biopsy was remarkable for inflammation
Line 376: Specify the homozygous frameshift mutation
Line 386: “pyridostigmine”; state the mechanism of action.
Figure 2: requires more detail (insets?) in order to capture the statements of the legend with respect to LAP1 interaction (influence of phosphorylation) with TRF2.
Line 17: Correct word “understating”
Line 21: Correct word “mutation” to mutations
